# Bedside Testing in Acute Vestibular Syndrome—Evaluating HINTS Plus and Beyond—A Critical Review

**Alexander A. Tarnutzer** [1,2,*] and **Jonathan A. Edlow** [3,4]

1   Department of Neurology, Cantonal Hospital of Baden, 5404 Baden, Switzerland
2   Faculty of Medicine, University of Zurich, 8006 Zurich, Switzerland
3   Emergency Medicine, Beth Israel Deaconess Medical Center, Boston, MA 02215, USA; jedlow@bidmc.harvard.edu
4   Emergency Medicine, Harvard Medical School, Boston, MA 02115, USA
*   Correspondence: alexander.tarnutzer@access.uzh.ch

**Abstract:** Acute vertigo and dizziness are frequent presenting symptoms in patients in the emergency department. These symptoms, which can be subtle and transient, present diagnostic challenges because they can be caused by a broad range of conditions that cut across many specialties and organ systems. Previous work has emphasized the value of combining structured history taking and a targeted examination focusing on subtle oculomotor signs. In this review, we discuss various diagnostic bedside algorithms proposed for the acutely dizzy patient. We analyzed these different approaches by calculating their area-under-the-curve (ROC) characteristics and sensitivity/specificity. We found that the algorithms that incorporated structured history taking and the use of subtle oculomotor signs had the highest diagnostic accuracy. In fact, both the HINTS+ bedside exam and the STANDING algorithm can more accurately diagnose acute strokes than early (<24 to 48 h after symptom onset) MRI with diffusion-weighted imaging (DWI). An important caveat is that HINTS and STANDING require moderate training to achieve this accuracy. Therefore, for physicians who have not undergone adequate training, other approaches are needed. These other approaches (e.g., ABCD2 score, PCI score, and TriAGe+ score) rely on vascular risk factors, clinical symptoms, and focal neurologic findings. While these other scores are easier for frontline providers to use, their diagnostic accuracy is far lower than HINTS+ or STANDING. Therefore, a focus on providing dedicated training in HINTS+ or STANDING techniques to frontline clinicians will be key to improving diagnostic accuracy and avoiding unnecessary brain imaging.

**Keywords:** vertigo; dizziness; bedside testing; HINTS; STANDING; truncal ataxia





## 1. Introduction

A chief complaint of acute vertigo or dizziness is related to about 2.1 to 7.1% of all emergency department (ED) visits [1–4], which have increased between 1995 and 2009 by 90% [5]. This translates to about 4.3 million ED visits in the USA per year as estimated in 2012 [6,7]. With the differential diagnosis of patients presenting with acute or episodic vertigo/dizziness being very broad and cutting across all specialties, frontline providers and specialists may be overwhelmed by the approach to this symptom. In patients presenting to the ED with vestibular symptoms, life-threatening conditions have been identified in a single study in 23.8% of visits, with 12.5% of all visits related to cerebrovascular events [2]. These investigators, from a Swiss tertiary-care hospital, defined an immediate "life-threatening" condition as that which "requires the highest priority medical assistance and often leads to admission to the intensive or intermediate care units or stroke unit." They also used a "modified version" of the dangerous diagnosis definition from another US study of a national database [3]. In the latter study, 15% of the 9472 patients identified had one of the pre-specified dangerous disorders, which included cerebrovascular and

cardiovascular disorders, as well as bacterial meningitis, fluid and electrolyte disturbances, hypoglycemia, and others.

This emphasizes the need to distinguish between dangerous central and benign, self-limited peripheral vestibular disorders to avoid misdiagnosis. Overall, roughly 9% of cerebrovascular events are missed at the initial ED presentation and risk of misdiagnosis is much greater when the presenting neurologic complaints are mild, nonspecific, or transient (range 24–60%) [8]. For posterior circulation strokes presenting with dizziness, frontline misdiagnosis appears common, occurring in roughly 35% of cases [9]. Thus, patients with stroke and vestibular neuritis can both present with an acute vestibular syndrome (AVS). Patients with AVS have acute-onset, continuous vertigo, dizziness, or unsteadiness lasting days to weeks, usually associated with vomiting, nystagmus, severe postural instability, and head movement intolerance [10].

Approximately 25% ± 15% of AVS patients will be diagnosed with a stroke, usually in the posterior circulation [11]. The desire to avoid missing a stroke often triggers brain imaging (computed tomography (CT) and magnetic resonance imaging including diffusion-weighted imaging (MRI-DWI)), laboratory workup, and/or electrocardiography. The annual costs in the USA related to the diagnosis and treatment of dizzy patients in the ED have been estimated to exceed USD 4 billion in 2013 and are expected to rise further [6,7]. Twelve percent of these costs are due to brain imaging [7]. The utilization of CT and MRI increased by 169% from 1995 to 2004, which was more than any other test [1]. Much more recent USA Medicare data reported that of 185,338 ED patients with dizziness, 46,852 (25%) had a CT, whereas only 6469 (4%) had an MRI in the ED [12]. Unfortunately, a recent meta-analysis (6 studies, 771 patients) found that the sensitivity of CT for central causes of dizziness was only 28.5% [13]. Negative CT results often reassure frontline clinicians that the patient does not have a central cause for their dizziness. This is best illustrated by a Canadian study, which reported that patients who were discharged from the ED with dizziness who had a CT during the ED visit were 2.3 times more likely to return with a stroke in the next 30 days compared to similar patients who had not had a CT [14].

Even MRIs including DWI in AVS have limited sensitivity, missing about one out of five vertebrobasilar strokes presenting as an AVS when obtained within the first 24–48 h after symptom onset [15]. This false-negative rate may rise to up to 50% for small lacunar brainstem strokes [16]. The underlying premise for all of this brain imaging is logical. Distinguishing a central from a peripheral cause of patients with AVS is crucial. Where a stroke occurs, the most common central cause of the AVS is missed or delayed, and the underlying stroke mechanism goes untreated, sometimes resulting in an extension of the original infarct or the development of a new, larger one.

Fortunately, the evidence base for effective bedside differentiation of inner ear diseases from stroke in patients with acute dizziness and vertigo has grown substantially over time [17,18], as emphasized recently [15]. Although useful if present, obvious focal neurologic signs only have a sensitivity of 44% for detecting a central cause of AVS, i.e., more than 50% of strokes will be missed if one were to rely on these findings [15]. Thus, different clinical strategies that emphasize the combined use of targeted neuro-otologic bedside examination techniques, such as HINTS (Head Impulse, Nystagmus, and Test of Skew) [19], HINTS+ (which adds a bedside test of hearing) [20], STANDING [21], or gait/truncal instability assessment, have been proposed [22].

Therefore, because of the limitations of current brain imaging techniques and because these various bedside evaluation tools have the potential to outperform imaging, we reviewed the diagnostic performance of these relatively new diagnostic algorithms for the diagnosis of patients with the AVS patient. We will discuss the advantages and limitations of these specific clinical tools.

## 2. Bedside Examination Tools in AVS

A broad range of clinical scores has been proposed to diagnose patients with AVS (see Table 1 for an overview). These scores utilize a combination of different features including

the patient's history (e.g., timing and triggers of the dizziness and cardiovascular risk factors), findings from the general neurologic examination, and findings from a dedicated oculomotor examination and biomarkers. The ABCD2 score, for example, relies on patient history and neurologic examination only [20,23,24], whereas the HINTS [19] and the STANDING [21] algorithms focus on findings from the dedicated oculomotor examination. Other scores combine the patient history, neurologic, and oculomotor examination (e.g., the TriAGe+ score) [25].

**Table 1.** Overview of proposed scores/algorithms for the assessment of the acutely dizzy patient.

| Score/Algorithm | General Clinical Elements Included | Specific Elements Tested | Evaluated Application | AUC (95% CI) | Sensitivity/Specificity (95% CI) * | Number of Studies Available, at Least One Validation Study Available (Yes/No) | Additional Training Required (Yes/No) | Advantages/ Disadvantages |
|---|---|---|---|---|---|---|---|---|
| HINTS [19] | Subtle oculomotor signs | Horizontal head-impulse test, horizontal gaze-evoked nystagmus, test of skew | AVS with nystagmus | 0.995 (0.985–1.000) [20] | 95.3% (92.5–98.1%)/92.6% (88.6–96.5%) [15] | Largest number of studies available (>10 LOE grade 1–3 studies). Validation studies available. | Yes, moderate training is needed (4–6 h [24,26]) for successful application. | High sensitivity and specificity. Only patients with at least one vascular risk factor included in original study [19]. |
| HINTS+ [20] | Subtle oculomotor signs | Horizontal head-impulse test, horizontal gaze-evoked nystagmus, test of skew, finger rub | AVS with nystagmus | NA | 97.2% (94.0–100.0%)/ 92.4% (86.9–97.9%) [15] | Large number of studies available (6 LOE grade 1–3 studies). Validation studies available. | Yes, moderate training is needed (4–6 h [24,26]) for successful application. | High sensitivity and specificity. Only patients with at least one vascular risk factor included in original study [19]. |
| STANDING [21,27] | Obvious focal neurologic signs and subtle oculomotor signs | Horizontal head-impulse test, horizontal gaze-evoked nystagmus, truncal ataxia, provocation maneuvers (Hallpike Dix, Pagnini–McClure) | Acute vertigo or dizziness | NA | 93.4–100%/ 71.8%–94.3% [28] | Moderate number of studies available, including 2 LOE grade 1–3 studies from one group). Internal and external validation available. | Yes, moderate training needed (4–6 h [24,26]) for successful application. | More inclusive than HINTS(+), covering positional vertigo (BPPV) also. |
| ABCD2 score [29] | Presenting sx, vascular risk factors, obvious focal neurologic signs | Age, blood pressure, clinical features (unilateral weakness, speech disturbance), duration of symptoms, diabetes | Acute vertigo or dizziness (some studies meeting criteria for AVS) | Range: 0.613 to 0.79 (0.61 (0.53–0.70) [20]; 0.69 (0.63–0.75) [30]; 0.73 (0.68–0.78) [25]; 0.79 (0.73–0.85) [29]) | For a cutoff value of ≥4: 55.7% (43.3–67.5%)/81.8% (76.4–86.2%) [24]; 61.1% (52–70%)/ 62.3% (51–72%) [20] | Moderate number of studies available, including 2 LOE grade 1–3 studies. Internal and external validation available. | No | Low diagnostic accuracy in acutely dizzy patients. Does not replace other scores such as HINTS or STANDING. |

**Table 1.** *Cont.*

| Score/Algorithm | General Clinical Elements Included | Specific Elements Tested | Evaluated Application | AUC (95% CI) | Sensitivity/Specificity (95% CI) * | Number of Studies Available, at Least One Validation Study Available (Yes/No) | Additional Training Required (Yes/No) | Advantages/Disadvantages |
|---|---|---|---|---|---|---|---|---|
| TriAGe+ score [25] | Presenting sx, vascular risk factors, obvious focal neurologic signs, subtle oculomotor signs | Triggers, atrial fibrillation, male gender, blood pressure ≥ 140/90 mm Hg, brain-stem/cerebellar dysfunction (incl. skew deviation, truncal ataxia), focal weakness or speech impairment, dizziness, no history of vertigo/dizziness, labyrinth/vestibular disease | Acute vertigo or dizziness | 0.82 (0.78–0.86) | For a cutoff value of 10 points: 77.5% (72.8–81.8%)/72.1% (64.1–79.2%), | Single center, retrospective study, with a single retrospective validation study that has serious limitations [31]. | No | Moderate diagnostic accuracy in acutely dizzy patients. Does not replace other scores such as HINTS or STANDING. |
| PCI score [30] | Past history, presenting sx, vascular risk factors, obvious focal neurologic signs | High blood pressure, diabetes mellitus, ischemic stroke, rotating and rocking, difficulty in speech, tinnitus, limb and sensory deficit, gait ataxia, and limb ataxia | Acute vertigo or dizziness | 0.82 (0.77 to 0.87) | For a cutoff value of 0 points: 94.1% (NA)/41.4% (NA) | Single center, retrospective study, no prospective validation studies available. | No | Moderate diagnostic accuracy in acutely dizzy patients (high sensitivity but low specificity). Does not replace other scores such as HINTS or STANDING. |

| Score/Algorithm | General Clinical Elements Included | Specific Elements Tested | Evaluated Application | AUC (95% CI) | Sensitivity/Specificity (95% CI) * | Number of Studies Available, at Least One Validation Study Available (Yes/No) | Additional Training Required (Yes/No) | Advantages/ Disadvantages |
|---|---|---|---|---|---|---|---|---|
| GTI rating [22,32–34] | Obvious focal neurologic signs | Gait and truncal instability (graded rating) | Acute vertigo, dizziness, or gait imbalance | NA | For a presence of truncal or gait ataxia: 69.7% (43.3–87.9%)/83.7% (52.1–96.0%) [28] | Moderate number of studies available, including 1 LOE 1 study [22]. Internal and external validation available. | No | Lower sensitivity than HINTS(+) or STANDING, but applicable also in patients with isolated truncal instability (without nystagmus) [34]. |

Abbreviations: AUC = area-under-curve values in receiver operating characteristics (ROC) curve; AVS = acute vestibular syndrome; BPPV = benign paroxysmal positional vertigo; CI = confidence interval; NA= not available; HINTS (Head impulse, nystagmus, test of skew); GTI = gait and truncal instability; LOE = level of evidence; PCI = posterior circulation infarct; STANDING = **S**pon**TA**neous and positional nystagmus, the evaluation of the **N**ystagmus **D**irection, the head impulse test, and the evaluation of equilibrium (standi**NG**); sx = symptoms. * Whenever available, data from systematic reviews and meta-analyses were reported.

### 2.1. HINTS/HINTS Plus

The three-component bedside HINTS (Head-Impulse, Nystagmus, and Test of Skew) can accurately identify central causes (mostly ischemic stroke) in AVS patients [19]. The three components include a bedside assessment of the horizontal angular vestibulo-ocular reflex (aVOR) by applying the head-impulse test (HIT) [35], evaluating ocular stability at eccentric gaze (looking for a gaze-evoked nystagmus) and testing for a vertical divergence in the alternating cover test (see Table 2). In the hands of a trained oto-neurologist, HINTS was associated with a 100% sensitivity and 96% specificity for detecting a stroke [19]. Importantly, the presence of one out of these three signs was sufficient to suspect a central cause. The HINTS paradigm has become increasingly popular since its introduction in 2009 and is now considered the standard bedside examination technique in AVS patients in the ED with the caveat that the examiner be trained in using HINTS.

**Table 2.** H.I.N.T.S. plus bedside testing battery * (modified after [36]).

| Test Performed | Property Evaluated | How to Perform This Test | Pointing to a Peripheral Cause | Pointing to a Central Cause | Comments |
|---|---|---|---|---|---|
| **Horizontal <u>H</u>ead-Impulse test (HIT)** | Vestibulo-ocular reflex (VOR) | Fast, low amplitude (10–15°) head rotations to the left/right while the patient is looking at a fixed target in space (e.g., the examiner's nose) | Delayed to one side, pathological catch-up saccade | Normal HIT. | Note that central lesions involving the VOR (e.g., lesions in the root-entry zone or of the vestibular nuclei) may show a "pseudo-peripheral pattern" |
| **Testing for <u>N</u>ystagmus** | Eccentric gaze-holding on lateral gaze | Fixation of an object (e.g., the tip of a pen) during lateral (eccentric) gaze (~20 to 30°) for at least 5 s. | Stable eccentric gaze-holding | Deficient eccentric gaze-holding with centripetal drift and centrifugal nystagmus (i.e., left-beating on left-gaze and right-beating on right-gaze). | Spontaneous, predominantly horizontal nystagmus (i.e., primary gaze nystagmus) can be found in both peripheral and central causes and thus allows no differentiation. |
| **Alternating cover test ("<u>T</u>est of <u>S</u>kew")** | Vertical alignment of the eyes | Rapid covering then uncovering one eye after the other while the patient is looking at a fixed target in space (e.g., the examiner's nose). The examiner should focus on only one eye. | No vertical deviation of the eyes | Vertical realignment of the uncovered eye (one eye goes up while the other eye goes down). This is why it does not matter which eye the examiner focuses on. | Note that rarely a vertical skew can also be observed in peripheral-vestibular deficits, but is usually of smaller amplitude and short-lived. |
| **New-onset unilateral hearing loss (fourth sign—"plus sign")** | Hearing | Finger rub on each side | Normal hearing | Hearing loss on the side with the abnormal head-impulse test | Hearing may also be compromised in inner ear disorders such as labyrinthitis or complicated otitis media, emphasizing the need for a dedicated examination of the ear. |

* Teaching videos can be found under: http://novel.utah.edu/Newman-Toker/collection.php (accessed on 15 August 2023).

This is evidenced by the recently published GRACE3 clinical guideline on acute vertigo and dizziness in the ED [37], which includes the specific recommendation that HINTS should be used in the ED only by trained clinicians. This is because current use by emergency clinicians in routine practice does not achieve the same results as those attained by trained subspecialists [38,39]. In a recent systematic review of the literature (1980–2022) focusing on high-quality (level of evidence 1 to 3) studies reporting on the diagnostic accuracy of bedside eye movement testing in acutely dizzy patients, ten studies investigating the diagnostic accuracy of bedside HINTS were included (representing data from 422 patients with a central AVS and 378 patients with a peripheral AVS). This meta-analysis reported a high sensitivity (95.3% [95% confidence interval (CI) = 92.5–98.1%]) and specificity (92.6% [88.6–96.5%]) for the HINTS bedside exam [15]. When adding a fourth sign (unilateral, new-onset hearing loss) to the HINTS battery (called HINTS-plus [20]), sensitivity increased further (97.2% [94.0—100.0%]) compared to the HINTS (by 1.9%). However, the sample size of patients was smaller for the HINTS+ battery (central AVS = 276, peripheral AVS = 252).

Importantly, both subspecialists (i.e., neuro-otologists/neuro-ophthalmologists) and *trained* non-subspecialists (i.e., general neurologists, neurology residents, and emergency physicians) demonstrated a high accuracy when using either the HINTS or the HINTS+ exam. Although the sensitivity of the HINTS exam was comparable amongst these two groups (94.3% vs. 95.0%, *p* = 0.55), the specificity of the HINTS exam was higher in the subspecialist group than in the non-subspecialist group (97.6% vs. 89.1%, *p* = 0.007), indicating potential differences in the interpretation of test results [15]. Another study, which fell outside of the inclusion criteria for this recent meta-analysis also suggested that trained ED clinicians can accurately perform and interpret two components of HINTS—the horizontal HIT and nystagmus testing [27]. Nevertheless, a limitation is that as of 2023, only small numbers of emergency clinicians have received adequate training in the HINTS exam. Moreover, the minimum effective curriculum, how to administer it, and its durability have not yet been clearly defined.

### 2.2. STANDING

The STANDING algorithm (i.e., a four-step algorithm including 1) the discrimination between **S**pon**TA**neous and positional nystagmus, (2) the evaluation of the **N**ystagmus **D**irection, (3) the head **I**mpulse test, and (4) the evaluation of equilibrium (sta**N**din**G**)) was designed to be more inclusive to include the diagnosis of benign paroxysmal positional vertigo (BPPV) as well. In addition to testing for spontaneous nystagmus, it also tests for positional nystagmus (by applying provocation maneuver for the posterior and lateral canals), and examines truncal ataxia [21,27]. However, this algorithm was more selective in applying single bedside tests based on initial findings (e.g., a head-impulse test was applied only in patients with unilateral spontaneous nystagmus). Likewise, the grading of truncal ataxia was less granular than proposed by others [22]. Specifically, an inability to stand or walk was considered indicative of a central origin, approximately reflecting grade 2 or grade 3 truncal instability (for details see the dedicated section on gait and truncal instability further below and in Table 3). It is important to note that the emergency physicians who participated in the STANDING trial all received training that included 4 h of lecture, 2 h of demonstration on normal volunteers, and 10 proctored exams on ED patients [27].

Three prospective studies with relatively unselected patient ED cohorts have been published. When first proposed, the developers of the STANDING algorithm reported a high overall diagnostic accuracy (sensitivity = 95% [83–99%]; specificity = 87% [85–87%]) in a cohort of 352 patients with acute vertigo/unsteadiness [27]. In a more recent, prospective study with 300 patients with isolated vertigo and unsteadiness, enrolled by a different group in a different country, the specificity of the STANDING algorithm was lower (75% vs. 87%), whereas the sensitivity (94% vs. 95%) was similar compared with the prior validation study [24]. In a follow-up prospective study, this second group investigated the

diagnostic accuracy of the STANDING algorithm performed by ED physicians (both interns and senior emergency physicians) who had received 4 h of training (lectures and practical demonstrations) [26]. The STANDING algorithm demonstrated sensitivities of 84.8% (75.6–93.9%) and 89.8% (82.1–97.5%) in the interns and the senior emergency physicians, respectively. Likewise, the specificity reached 88.9% (85.1–92.8%) and 91.3% (87.8–94.8%) in both respective groups.

**Table 3.** Graded rating of gait and truncal instability (GTI).

| Grade of Gait Inability | Definition |
| --- | --- |
| 0 | Normal gait |
| 1 | Mild to moderate imbalance but can walk independently [32], or unable to stand on tandem Romberg with the eyes open at least for 3 s [33]. |
| 2 | Severe imbalance with standing and cannot walk without support [32], or unable to stand on tandem Romberg with eyes open for 3 s [33]. |
| 3 | Inability to stand upright unassisted [32,33], or inability to sit upright unassisted [33]. |

Overall, these prospective studies confirm that the STANDING algorithm is valuable in the ED setting, with a diagnostic accuracy similar to that when using the HINTS. Advantages include the ability to diagnose BPPV, of both the posterior and lateral canals, which is far more common than posterior circulation stroke presenting as isolated dizziness. A confident diagnosis of a peripheral problem makes a central cause extremely unlikely. Another advantage is that STANDING is "blind" to the presenting timing and triggers. Because some patients with BPPV will present early and have lingering symptoms, mimicking an AVS [40], STANDING can be used in these patients too.

A potential limitation is the relatively small number of studies published, with two of the three studies available coming from the same group. Furthermore, all of the emergency physicians involved received training and used Frenzel lenses, which is not standard practice for emergency physicians.

### 2.3. TriAGe+ Score and PCI-Score

The TriAGe+ score incorporates information from the patient's history (the triggers and the type of dizziness and the presence or absence of vascular risk factors) and from the bedside physical examination (including performing the alternating cover test and testing of stance and gait), which are combined. This score (range = 0–17 points) was first proposed by Kuroda and co-workers in 2017 and was compared to the ABCD2 score [25]. In a single-center observational retrospective study, 498 patients presenting to the ED with vertigo or dizziness were included [25]. Based on the area under the curve (AUC), the diagnostic accuracy of the TriAGe+ score was maximal when selecting a cutoff value of 10 points, resulting in a sensitivity of 77.5% and a specificity of 72.1%. Compared to the ABCD2 score, the TriAGe+ score had a significantly larger AUC for the occurrence of stroke ($p < 0.001$), although well below diagnostic accuracy values reported for the HINTS(+) or STANDING algorithms.

Recently, a single-center, retrospective validation study of the TriAGE+ score of 444 ED patients with dizziness, of whom 73 (16.4%) had strokes, was published [31]. This study has two important limitations—some patients had findings beyond isolated dizziness (e.g., brainstem findings, facial palsy, aphasia, and "cerebellar findings"), and they only included patients who had an MRI as part of their routine care, both of which could affect their results. Nevertheless, their findings were largely in line with those of Kuroda; they found that when using a cutoff of ≥5, the TriAGE+ score was 100% sensitive but only a 16% specificity for stroke. Notably, the HIT and nystagmus testing on the lateral-gaze test are

not part of the TriAGe+ score and more detailed information on how truncal instability was rated is missing. Considering this evidence as a whole, the value of the TriAGe+ score currently remains unclear, and better-validated scores such as HINTS+ or STANDING perform better.

Likewise, the PCI score combines nine items, addressing reported symptoms (type of dizziness), vascular risk factors, and focal neurologic findings such as limb or gait ataxia [30]. Importantly, the PCI score also does not include any subtle oculomotor signs. As with the TriAGe+ score, the AUC for the PCI score was significantly larger than for the ABCD2 score (0.82 vs. 0.69), and has a sensitivity of 94.1% and a specificity of 41.4%. Importantly, this score was retrieved from a retrospective data set and was not prospectively validated, substantially limiting its current clinical applicability.

### 2.4. ABCD2 Score

The ABCD2 score (0–7 points) was originally developed as an epidemiologic tool to predict the stroke risk in patients after a transient-ischemic attack (TIA) [41]. In a retrospective study by Navi and colleagues, it has been suggested that such a risk stratification approach based on the ABCD2 score might help identify strokes acutely in ED patients presenting with dizziness [29]. Being based on five items readily assessable in the ED setting, it can be reliably and quickly calculated by emergency clinicians. Navi and colleagues identified a cutoff value of four points or more as an indication of a central (usually ischemic) cause. The diagnostic accuracy of the ABCD2 score has been compared to other algorithms proposed for distinguishing peripheral from central causes in acutely dizzy patients, including the TriAGe+ score, the PCI score, the HINTS+ bedside exam, and the STANDING algorithm.

In a prospective, cross-sectional study including high-risk patients with AVS (*n* = 190), using brain MRI including DWI as the gold standard in all patients, the AUC of the ROC curve was significantly smaller for the ABCD2 score (0.613, [0.531–0.695]) than for HINTS (0.995 [0.985–1.000]) [20]. Thus, HINTS (stroke sensitivity = 96.5%, specificity = 84.4%) substantially outperformed the ABCD2 score (cutoff value= $\geq$ 4 points, sensitivity = 61.1%, specificity = 62.3%) for stroke diagnosis in ED patients with AVS. Another prospective study compared the HINTS, STANDING, and ABCD2 scores in a single-center diagnostic cohort study among patients with isolated vertigo and unsteadiness presenting to the ED [24]. Both the HINTS and the STANDING algorithms reached high sensitivities of 97% and 94% and negative predictive values (NVP) of 99% and 98%, respectively. However, the ABCD2 score failed to predict half of the central vertigo cases and had a sensitivity of 55% and an NPV of 87% [24]. Likewise, the ABCD2 score was inferior to the TriAGe+ score [25] and the PCI score [30] as described above. A prospective, single-center cross-sectional study including patients with acute dizziness presenting to the ED, compared the diagnostic accuracy of the ABCD2 score and HINTS [23]. All patients received a brain MRI including DWI at least 48 h after symptom onset. Whereas the sensitivity of the ABCD2 score for stroke was 71.4% for a score of $\geq$4, these authors reported 100% sensitivity for the HINTS exam. Interestingly, when using a combination of a "central pattern of nystagmus", defined as the presence of a bidirectional gaze-evoked nystagmus, isolated torsional nystagmus, or vertical nystagmus in any position, plus an ABCD2 score of $\geq$4, a sensitivity for detecting central causes of 100% was achieved as well.

### 2.5. Gait and Truncal Instability (GTI) Rating

In the hands of neuro-otologists and trained ED physicians, HINTS(+) has been very successfully applied [15]. Less experienced or untrained emergency physicians, do not use HINTS(+) properly, either using them on the wrong patients, performing the test improperly, or interpreting the results incorrectly [38,39]. Until training of this group is successfully implemented at scale, other accurate tests that do not rely on subtle oculomotor findings might help. Gait assessment is an established part of the basic standard ED neurological exam for a dizzy patient. In addition, knowing whether or not a patient has a

safe and stable gait is an important element of a safe discharge for ED patients, no matter what the cause. Finally, an inability to walk independently would strongly favor a central cause of dizziness and should make the clinician question very common diagnoses such as BPPV [40].

For these reasons, assessing for gait and truncal instability (GTI) has been proposed as a substitute for the HINTS exam for ED physicians who have not received any training in performing HINTS [22]. In an attempt to provide a graded truncal instability rating, different clinical findings have been linked to grade 1, 2, and 3 truncal instability (see Table 3).

In a recently published meta-analysis including ten studies reporting on GTI in acutely dizzy patients, pooled estimated sensitivity reached 69.7% (43.3–87.9%) and specificity was at 83.7% (52.1–96.0%) when considering GTI ratings of two or three as indicative of a central cause [28]. When comparing the performance of ED physicians and neurologists, a low correlation (Spearman's correlation $r^2 = 0.17$) was reported in a single study [42]. The investigators did not speculate on the reason for this disparity, nor did they report details about how the disparities might have affected patients' management. This makes it difficult to account for this finding.

Focusing on grade 3 GTI, another recent meta-analysis found a sensitivity of 35.8% (5.2–66.5%) and a specificity of 99.2% (97.8–100.0%), emphasizing that the presence of grade 3 GTI is highly suggestive of a central cause [15], whereas the absence of grade 3 GTI does not exclude presence of a central cause of AVS (missing 2/3 of all vertebrobasilar strokes with this cutoff value). Furthermore, in patients presenting with acute truncal ataxia without (spontaneous or gaze-evoked) nystagmus, HINTS may not be applicable. Considering the graded GTI rating instead may therefore be useful, as recently demonstrated by Carmona and colleagues [34].

Importantly, several limiting factors in GTI analysis need to be considered. First, the timing of GTI testing varied among studies. Whereas some studies applied truncal instability testing early in the clinical examination, others performed testing only after having the patient rest for at least 5–10 min. Secondly, severe nausea or motion intolerance may prevent adequate testing for grades 2 and 3 GTI [43]. However, assessing the ability to sit up on the stretcher without holding on to the guard rails can be considered a proxy for the GTI assessment, allowing detection of those patients with severe (i.e., grade 3) gait and truncal instability who are very likely to have a central cause of their AVS.

## 3. Discussion

Missed or delayed diagnosis of posterior circulation stroke is an important and, unfortunately, a common problem [8,44–46]. This can lead to serious negative health outcomes, chief among which is an extension of the initial stroke or the development of a second one with more significant clinical deficits or death [47].

For the bedside clinical assessment of acutely dizzy patients meeting the diagnostic criteria of an acute vestibular syndrome, looking for subtle oculomotor signs is key to increasing diagnostic accuracy. This approach was initiated by the introduction of the HINTS examination in 2009, with growing popularity among specialists. With more than ten high-level-of-evidence studies reporting on unselected patient populations, the utility of the HINTS, when performed by trained examiners, is clear. Thus, it is not surprising that the GRACE-3 clinical guideline, created by emergency physicians strongly recommends performing the bedside HINTS to optimally manage ED patients with acute dizziness [37]. We want to reiterate that the GRACE-3 guideline also included a strong mandate for training in order for the use of HINTS to be implemented effectively.

Both the HINTS(+) exam and the STANDING algorithm are very good exclusion tests in the hands of ***trained*** emergency physicians, non-sub-specialists, and neuro-otology/ neuro-ophthalmology subspecialists [15]. Gerlier and colleagues reported that as few as six hours of training by an otologist was sufficient to reliably perform and interpret both HINTS and STANDING algorithms [24]. It is not known if further training would yield similar

results. Thus, in the right circumstances, both HINTS and the STANDING algorithm can distinguish peripheral from central diagnoses, limiting the use of further diagnostic testing to cases where bedside testing points to a central (or equivocal) cause of AVS. Ideally, HINTS+ are combined with a graded GTI rating or the STANDING algorithm is used instead. Although gait testing is an integral and important part of the clinical evaluation, especially if the patient is unable to walk independently, using it as a stand-alone test is less diagnostically accurate than HINTS+ and STANDING. Other scores or grading systems reviewed here are less accurate and lack high-quality, prospective validation studies. This is true for the ABCD2 score, the TriAGe+ score, and the PCI score. The ABCD2 score was never intended to be used to distinguish peripheral from central causes of dizziness. Both the TriAGe+ and PCI scores are intended for that purpose but, to some extent, were designed as a "work-around" because most non-sub-specialists are not trained in evaluating subtle oculomotor findings. However, directly assessing the neurophysiology by direct physical examination will always trump epidemiological context. Application of these other scoring systems in AVS patients cannot be recommended at this time.

Importantly, both HINTS(+) [15] and STANDING outperformed early (i.e., within the first 24–48 h) MRI-DWI, which has a sensitivity of 81.1 (73.3–88.8) and a specificity of 99.9 (99.6–100.0) (based on a systematic review from [13]), as shown in Figure 1. Likewise, the GTI rating outperformed brain CT [15]. This is expected since the same systematic review found a sensitivity of CT for central causes of dizziness to be less than 30% [40]. For small brainstem strokes (with a diameter of $\leq 10$ mm), the sensitivity of early MRI-DWI may be as low as 47% [16], and nearly half of these strokes were due to large vessel pathology. This underscores the importance of bedside oculomotor testing in the management of acutely dizzy patients.

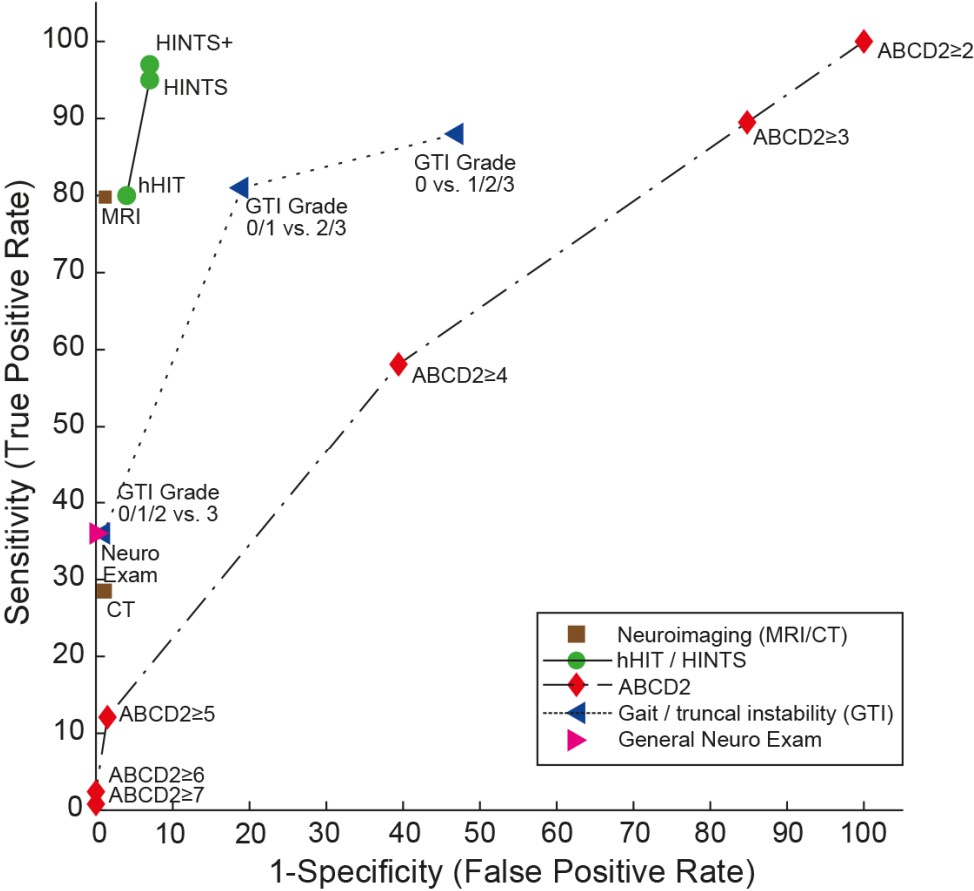

**Figure 1.** Summary receiver operating characteristic (SROC) curve analysis for the "HINTS (Head Impulse, Nystagmus, Test of Skew) family" compared with neuroimaging (computed tomography (CT)

or magnetic resonance imaging with diffusion-weighted sequences (MRI-DWI), values used as published by Shah and colleagues [13]), graded gait/truncal instability (GTI) ratings, general neurologic exam, and vascular risk stratification by ABCD2 (age, blood pressure, clinical features, duration of symptoms, diabetes) score (data from a single study [20]) for detecting stroke in patients presenting the acute vestibular syndrome (modified after [48]). SROC curves are shown for five different diagnostic approaches to diagnosing stroke in the acute vestibular syndrome. A perfect test or decision rule has threshold cutoffs in the upper left corner (100% sensitivity, 100% specificity) and an area under the curve (AUC) of 1.0. Note that the gait/truncal instability ratings outperform the ABCD2 score and the general neurologic exam but are clearly inferior compared to the HINTS family of eye movement tests. Both HINTS and HINTS plus (HINTS plus new hearing loss detected by finger rubbing or similar) demonstrate a higher diagnostic accuracy for ruling out stroke than MRI including DWI. Reused with permission from John Wiley and Sons. Abbreviations: hHIT = horizontal head-impulse test.

## 4. Future Directions

The diagnostic work-up of the acutely dizzy patient remains challenging, and absent or inadequate training in performing and interpreting subtle oculomotor findings constitutes important limitations, especially in frontline providers. Most ED physicians are not yet familiar with a structured approach to the dizzy patient as, e.g., outlined in the TiTrATE approach and HINTS(+) or similar bedside algorithms [18,37].

While the introduction of quantitative (video oculography—VOG) HINTS to the ED seems promising, increasing the diagnostic accuracy beyond that of neuro-otology experts [49,50], this concept is far from being implemented broadly in routine emergency medicine practice. Significant obstacles (availability of expensive equipment and the means to interpret the findings) exist. That said, over time, VOG could become an important quality assurance tool so that frontline providers could have their results "over-read" by specialists to increase their proficiency.

There is tempered validation of the findings of Wang et al. [22], that the combination of a central pattern of nystagmus plus an ABCD2 score ≥ 4 being 100% sensitive to stroke might be an effective strategy for frontline clinicians who have not mastered the HIT but are able to learn to identify central nystagmus. However, given that half of patients with cerebellar stroke do not exhibit nystagmus, we remain cautious about this approach [51].

## 5. Conclusions

Disseminating knowledge about the management of the acutely dizzy patient to frontline providers and providing dedicated training in HINTS+ or STANDING techniques will remain key to improving diagnostic accuracy and avoiding unnecessary brain imaging.

**Author Contributions:** A.A.T. was responsible for all contents of this manuscript, including conceptualization of the critical review, reviewing identified manuscripts, drafting and editing the manuscript; A.A.T. approved the final version of the manuscript; J.A.E. made substantial edits to the manuscript. All authors have read and agreed to the published version of the manuscript.

**Funding:** This research received no external funding.

**Institutional Review Board Statement:** Not applicable.

**Informed Consent Statement:** Not applicable.

**Data Availability Statement:** No new data were created or analyzed in this study. Data sharing is not applicable to this article.

**Conflicts of Interest:** One author (A.A.T.) declares no conflict of interest. The other author (J.A.E.) reviews cases involving medical malpractice for both plaintiff and defense firms.

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
