# Peer review of "Bedside Testing in Acute Vestibular Syndrome—Evaluating HINTS Plus and Beyond—A Critical Review"

_audiolres, doi:10.3390/audiolres13050059_

Round 1

Reviewer 1 Report

This article reviewed multiple diagnostic criteria for the acute vestibular syndrome, and showed that HINTS+ is very useful in assessing vestibular and neurological symptom and signs. The content includes well-researched reviews of each diagnostic criteria and will be valuable for clinicians.

Author Response

Reply by the author: We thank the reviewer for this positive review.

Reviewer 2 Report

Manuscript ID

Audiolres-2477520

Study Title: Bedside testing in acute vestibular syndrome—HINTS plus and beyond—A critical review

Reviewer’s Comments to Authors

Thanks for providing the opportunity to review such an interesting manuscript focused on Bedside testing in acute vestibular syndrome—HINTS plus and beyond—A critical review. It is an interesting and novel study that will add value to the scientific world and physiotherapy practices. The manuscript has been written well and concisely presented for easy reading. However, I would suggest some general and specific comments to make this manuscript the best version to be published if accepted.

General comments

Title: "Bedside Testing in Acute Vestibular Syndrome—Evaluating HINTS Plus and Beyond: A Critical Review" (Adding "evaluating" gives the reader a clear idea of what the review is about

Abstract

Line 10: Rephrase this sentence for better clarity "Acute vertigo and dizziness frequently present as symptoms in emergency department patients." 

Line 11-12: Rearrange this sentence "The broad differential diagnosis and often subtle and transient symptoms/signs make the diagnostic approach to dizzy patients challenging."

Line 13-14: Add the purpose of the review firstly "This review discusses various bedside algorithms proposed for diagnosing acutely dizzy patients, emphasizing the value of combining structured history-taking with a targeted examination focusing on subtle oculomotor signs."

Line 15-17: Break down the sentence for better readability "Comparing ROC-characteristics, sensitivity, and specificity, it is observed that algorithms focusing on subtle oculomotor signs exhibit the highest diagnostic accuracy. Specifically, the HINTS+ bedside-exam and the STANDING algorithm have demonstrated higher diagnostic accuracy for identifying strokes than early MRI-DWI (conducted <24-48h after symptom-onset)."

Line 18-20: "However, these require moderate training for reliable application, suggesting the need for other approaches for physicians lacking this specialized training." (Making it more straightforward)

Line 21-23: Break down long sentence "Several scores that depend on vascular risk-factors, clinical symptoms, and focal neurological findings, such as the ABCD2-score, PCI-score, and TriAGe+-score, are more easily applied by frontline providers. However, they provide only intermediate diagnostic accuracy and often lack external validation."

Line 24-26: Make this sentence more precise "These scores cannot substitute algorithms that assess subtle oculomotor findings. It is essential to provide dedicated training in HINTS+ or STANDING techniques to improve diagnostic accuracy and avoid unnecessary brain imaging."

Introduction:

Line 36-38: Provide more information or context about the referenced study that found life-threatening conditions in 23.8% of visits.

Line 48-49: Consider simplifying the definition of AVS, it's currently quite complex and may be difficult for some readers to comprehend.

Line 50-54: Make clear the connection between the diagnosis of vertebrobasilar ischemic stroke and the subsequent testing procedures (CT and MRI) and costs.

Line 64-65: Define DWI (diffusion-weighted imaging) before it is first used to make sure readers understand what it is.

Line 71-73: Mention why the detection of a central cause of AVS is crucial.

Line 76: The transition to the introduction of the review seems abrupt. Consider providing a more detailed explanation of why it is essential to have diagnostic algorithms for AVS.

Line 103: The mention of the GRACE3-guideline could benefit from more context. Consider providing a brief explanation of what this guideline is.

Line 203: A minor typo, the term should be "negative predictive value (NPV)" instead of "negative predictive values (NVP)".

Line 218-219: The phrase "less demanding bedside tests" could be misinterpreted. Clarify whether "less demanding" refers to the simplicity of execution, less time-consuming nature, or a lesser degree of expertise required.

Line 224: Consider rephrasing this to clearly indicate whether the substitution of the GTI for the HINTS exam is recommended or it is just a measure taken in cases where there's lack of training.

Line 226-232: A table or a diagram illustrating the grading of truncal instability would be helpful for readers to visualize and understand the concept better.

Line 237-238: The correlation between the performance of ED physicians and neurologists in using the GTI ratings is reported to be low. It might be helpful to discuss or speculate why this might be the case, in order to give the reader a better understanding of the context.

Overall, this table is well-structured and contains a lot of relevant information, which is good. However, there are a few areas where additional clarification or adjustment might make it easier for the reader to understand and interpret the table.

Header row: The term "Domains tested" could be clarified. Does this refer to the areas of symptomatology, the aspects of patient history, or the physical examination components that are evaluated?

The "Features evaluated" column: It might be clearer to split this into two separate columns, one for the specific tests performed and another for the features evaluated by these tests.

The column of "Comments": This column could potentially benefit from standardizing the type of information provided. For instance, some comments refer to the number of studies, some to the training needed, and some to the population included in the studies. If possible, provide the same type of information for each algorithm.

TriAGe+ score and PCI score rows: Note that these are based on single center, retrospective studies. It might be worth noting the limitations of this kind of study design, as this may affect the generalizability of the scores.

GTI rating row: Mention of the "graded rating" without context or a reference to look up further details may confuse readers. Consider adding a brief explanation or reference here.

Discussion:

 Line 34-36: The argument presented here could be stronger if it included more specific data to support the claim that standalone use of the GTI rating results in inferior diagnostic accuracy.

Line 38-40: Although the ABCD2-score, the TriAGe+ score, and the PCI-score are dismissed due to lack of high-quality, prospective validation studies, the authors might want to provide some context or additional explanation for why these other scoring systems were developed and are still in use.

Line 43-45: The comparison of the HINTS(+) and STANDING tests with MRI-DWI and brain CT is interesting, but it would be helpful to see more discussion on what this means in practical terms for clinicians and patients.

Line 50-56: The discussion on the challenges of diagnosing acute dizziness could be expanded to provide a more complete picture of these difficulties. For instance, discussion about how these difficulties might affect patient outcomes would be beneficial.

Line 57-61: While the authors mention the potential of quantitative HINTS and the obstacles to their wider implementation, more context or examples would help underscore the importance of this point.

For a well-rounded discussion, it would be advantageous to include some recommendations for future research or suggestions for how to address the limitations and challenges identified.

Add conclusion

Author Response

Thanks for providing the opportunity to review such an interesting manuscript focused on Bedside testing in acute vestibular syndrome—HINTS plus and beyond—A critical review. It is an interesting and novel study that will add value to the scientific world and physiotherapy practices. The manuscript has been written well and concisely presented for easy reading. However, I would suggest some general and specific comments to make this manuscript the best version to be published if accepted.

Reply by the author: We thank the reviewer for this positive review and the suggestions made for improvement. A point-to-point reply to all comments is provided below.

General comments

Title: "Bedside Testing in Acute Vestibular Syndrome—Evaluating HINTS Plus and Beyond: A Critical Review" (Adding "evaluating" gives the reader a clear idea of what the review is about.

Reply by the author: We agree that this is reasonable and have added it to the title.

Abstract

Line 10: Rephrase this sentence for better clarity "Acute vertigo and dizziness frequently present as symptoms in emergency department patients." 

Reply by the author:

Agree and done. It now reads: “Acute vertigo and dizziness are frequent presenting symptoms in patients in the emergency department”

Line 11-12: Rearrange this sentence "The broad differential diagnosis and often subtle and transient symptoms/signs make the diagnostic approach to dizzy patients challenging."

Reply by the author:

Agree and done. It now reads: “These symptoms, which can be subtle and transient, present diagnostic challenges because they can be caused by a broad range of conditions that cut across many specialties and organ systems.”

Line 13-14: Add the purpose of the review firstly "This review discusses various bedside algorithms proposed for diagnosing acutely dizzy patients, emphasizing the value of combining structured history-taking with a targeted examination focusing on subtle oculomotor signs."

Reply by the author:

Agree and re-organized as suggested.

Line 15-17: Break down the sentence for better readability "Comparing ROC-characteristics, sensitivity, and specificity, it is observed that algorithms focusing on subtle oculomotor signs exhibit the highest diagnostic accuracy. Specifically, the HINTS+ bedside-exam and the STANDING algorithm have demonstrated higher diagnostic accuracy for identifying strokes than early MRI-DWI (conducted <24-48h after symptom-onset)."

Reply by the author:

Agree and re-worded. It now reads: “We analyzed these different approaches by calculating their area-under-the-curve (ROC) characteristics and sensitivity/specificity. We found that the algorithms that incorporated structured history-taking and the use of subtle oculomotor signs had the highest diagnostic accuracy. In fact, both the HINTS+ bedside-exam and the STANDING algorithm can more accurately diagnose acute strokes than early (<24-48h after symptom-onset) MRI-with diffusion weighted imaging (DWI).”

Line 18-20: "However, these require moderate training for reliable application, suggesting the need for other approaches for physicians lacking this specialized training." (Making it more straightforward)

Reply by the author:

Agree and re-worded and re-organized. It now reads: “An important caveat is that HINTS and STANDING require moderate training to achieve this accuracy. Therefore, for physicians who have not undergone adequate training, other approaches are needed.”

Line 21-23: Break down long sentence "Several scores that depend on vascular risk-factors, clinical symptoms, and focal neurological findings, such as the ABCD2-score, PCI-score, and TriAGe+-score, are more easily applied by frontline providers. However, they provide only intermediate diagnostic accuracy and often lack external validation."

Reply by the author:

Agree and done. It now reads: “These other approaches (e.g., ABCD2 score, PCI score, TriAGe+-score) rely on vascular risk-factors, clinical symptoms and focal neurologic findings. While these other scores are easier for frontline providers to use, their diagnostic accuracy is far lower than HINTS+ or STANDING.”

Line 24-26: Make this sentence more precise "These scores cannot substitute algorithms that assess subtle oculomotor findings. It is essential to provide dedicated training in HINTS+ or STANDING techniques to improve diagnostic accuracy and avoid unnecessary brain imaging."

Reply by the author:

Agree and re-worded. It now reads: “While these other scores are easier for frontline providers to use, their diagnostic accuracy is far lower than HINTS+ or STANDING.  Therefore, focus on providing dedicated training in HINTS+ or STANDING techniques to frontline clinicians will be key to improve the diagnostic accuracy and to avoid unnecessary brain imaging.”

Introduction:

Line 36-38: Provide more information or context about the referenced study that found life-threatening conditions in 23.8% of visits.

Reply by the author:

We fleshed out details that explain how these authors defined “life-threatening” and added some language from Reference #3, to which Reference #2 refers.

It now reads: “In patients presenting to the ED with vestibular symptoms, life-threatening conditions have been identified in a single study in 23.8% of visits, with 12.5% of all visits related to cerebrovascular events [2]. These investigators, from a Swiss tertiary care hospital, defined immediately “life-threatening” conditions as those which “requires the highest priority medical assistance and often leads to admission to the intensive or intermediate care units or stroke unit.” They also used a “modified version” of the dangerous diagnosis definition from another US study of a national database [3]. In the latter study, 15% of 9472 patients identified had one of the pre-specified dangerous disorders, which included cerebrovascular and cardiovascular disorders as well as bacterial meningitis, fluid and electrolyte disturbances, hypoglycemia and others.”

Line 48-49: Consider simplifying the definition of AVS, it's currently quite complex and may be difficult for some readers to comprehend.

Reply by the author:

We have simplified the sentence structure of this definition but we do not think that it can be significantly shortened without losing some important aspects of the definition. We feel that the new wording is easier to understand and will hopefully mitigate this specific comment.

It now reads: “Patients with the AVS have acute-onset, continuous vertigo, dizziness, or unsteadiness lasting days to weeks, usually associated with vomiting, nystagmus, severe postural instability and head movement intolerance [10].”

Line 50-54: Make clear the connection between the diagnosis of vertebrobasilar ischemic stroke and the subsequent testing procedures (CT and MRI) and costs.

Reply by the author:

We have substantially reorganized this section which we believe makes the points more clearly.

This section now reads: “The desire to avoid missing a stroke often triggers brain imaging (computed tomography (CT) and magnetic resonance imaging including diffusion-weighted imaging (MRI-DWI)), laboratory workup and/or electrocardiography. The annual costs in the USA related to the diagnosis and treatment of dizzy patients in the ED have been estimated to exceed 4 billion USD in 2013 and are expected to rise further [6,7]. Twelve percent of these costs are due to brain imaging [7]. The utilization of CT and MRI increased by 169% from 1995 to 2004, which was more than any other test [1]. Much more recent USA Medicare data reported that of 185,338 ED patients with dizziness, 46,852 (25%) had a CT whereas only 6469 (4%) had an MRI in the ED [12]. Unfortunately, a recent meta-analysis (6 studies, 771 patients) found that the sensitivity of CT for central causes of dizziness was only 28.5% [13].  Negative CT results often reassure frontline clinicians that the patient does not have a central cause for their dizziness. This is best illustrated by a Canadian study, which reported that patients who were discharged from the ED with dizziness who had a CT during the ED visit were 2.3 times more likely to return with a stroke in the next 30 days compared to similar patients who had not had a CT [14].

Even MRI including DWI in AVS has limited sensitivity, missing about 1 out of 5 vertebrobasilar strokes presenting as an AVS when obtained within the first 24-48 hours after symptom onset [15]. This false-negative rate may rise to up to 50% for small lacunar brainstem strokes [16]. The underlying premise for all of this brain imaging is logical. Distinguishing a central from a peripheral cause of patients with the AVS is crucial. If stroke, the most common central cause of the AVS is missed or delayed, the underlying stroke mechanism goes untreated sometimes resulting in extension of the original infarct or development of a new, larger one.”

Line 64-65: Define DWI (diffusion-weighted imaging) before it is first used to make sure readers understand what it is.

Reply by the author:

The first instance actually occurs higher up in the article, so we made the change there.

Line 71-73: Mention why the detection of a central cause of AVS is crucial.

Reply by the author:

We did this in the paragraph that begins talking about MRI sensitivity.

Line 76: The transition to the introduction of the review seems abrupt. Consider providing a more detailed explanation of why it is essential to have diagnostic algorithms for AVS.

Reply by the author:

We softened this transition. It now reads: “Therefore, because of the limitations of current brain imaging techniques and because these various bedside evaluation tools have the potential to outperform imaging, we reviewed the diagnostic performance of these relatively new diagnostic algorithms for diagnosis of patients with the AVS patient. We will discuss advantages and limitations of these specific clinical tools.”

Line 103: The mention of the GRACE3-guideline could benefit from more context. Consider providing a brief explanation of what this guideline is.

Reply by the author:

Agree and done. It now reads: “This is evidenced by the recently published GRACE3 clinical guideline on acute vertigo and dizziness in the ED, [27]. which includes the specific recommendation that HINTS should be used in the ED only by trained clinicians. This is because current use by emergency clinicians in routine practice does not achieve the same results as those attained by trained subspecialists [28,29].”

Line 203: A minor typo, the term should be "negative predictive value (NPV)" instead of "negative predictive values (NVP)".

Reply by the author:

Actually, the plural is correct as we are referring to 2 studies.

Line 218-219: The phrase "less demanding bedside tests" could be misinterpreted. Clarify whether "less demanding" refers to the simplicity of execution, less time-consuming nature, or a lesser degree of expertise required.

Reply by the author:

Agree and we changed the language. It now reads: “Until training of this group is successfully implemented at scale, other accurate tests that do not rely on subtle oculomotor findings might help.”

Line 224: Consider rephrasing this to clearly indicate whether the substitution of the GTI for the HINTS exam is recommended or it is just a measure taken in cases where there's lack of training.

Reply by the author:

We have changed this language and added other reasons why gait testing is important. It now reads: “Gait assessment is an established part of the basic standard ED neurological exam for a dizzy patient. In addition, knowing whether or not a patient has a safe and stable gait is an important element of a safe discharge for ED patients, no matter what the cause. Finally, inability to walk independently would strongly favor a central cause of dizziness and should make the clinician question very common diagnoses such as BPPV [30].”

Line 226-232: A table or a diagram illustrating the grading of truncal instability would be helpful for readers to visualize and understand the concept better.

Reply by the author:

Good idea and table made (Table 3 in the revised manuscript).

Line 237-238: The correlation between the performance of ED physicians and neurologists in using the GTI ratings is reported to be low. It might be helpful to discuss or speculate why this might be the case, in order to give the reader a better understanding of the context.

Reply by the author:

It’s hard to speculate on this given that article did not present any details on this. We added this language. It now reads: “When comparing performance of ED physicians and neurologists, a low correlation (Spearman’s correlation r2=0.17) was reported in a single study [38]. The investigators did not speculate on the reason for this disparity, nor did they report details about how the disparities might have affected patients’ management. This makes is difficult to account for this finding.”

Another potential fix is to eliminate the sentence, which we can do if the editor requests it.

Overall, this table is well-structured and contains a lot of relevant information, which is good. However, there are a few areas where additional clarification or adjustment might make it easier for the reader to understand and interpret the table.

Reply by the author:

We have reviewed the table (Table 1) and made adjustments to the headers and added more detail to comments made.

Header row: The term "Domains tested" could be clarified. Does this refer to the areas of symptomatology, the aspects of patient history, or the physical examination components that are evaluated?

Reply by the author:

We changed the wording to clarify. It now reads: “General clinical elements included”

The "Features evaluated" column: It might be clearer to split this into two separate columns, one for the specific tests performed and another for the features evaluated by these tests.

Reply by the author:

We feel that for this audience, this information is already known. Furthermore, we do provide detailed information on key oculomotor tests performed in the HINTS testing battery in table 3.

The column of "Comments": This column could potentially benefit from standardizing the type of information provided. For instance, some comments refer to the number of studies, some to the training needed, and some to the population included in the studies. If possible, provide the same type of information for each algorithm.

Reply by the author:

We thank the reviewer for this helpful input. We have split up the “comments” column in three columns, entitled “Number of studies available, at least one validation study available (yes/no)”, “Additional training required (yes/no)” and “Advantages / disadvantages” and now provide information to these aspects for all scores listed.

TriAGe+ score and PCI score rows: Note that these are based on single center, retrospective studies. It might be worth noting the limitations of this kind of study design, as this may affect the generalizability of the scores.

Reply by the author:

We have added an “advantages / disadvantages” column in the table, addressing this limitation now.

GTI rating row: Mention of the "graded rating" without context or a reference to look up further details may confuse readers. Consider adding a brief explanation or reference here.

Reply by the author:

We disagree with this comment. We have already clarified the GTI rating in a new Table (Table 3). Presenting the information on GTI degree is parallel to presenting ABCD2 by degree/score. We do not feel this change would be more clear.

Discussion:

Line 34-36: The argument presented here could be stronger if it included more specific data to support the claim that standalone use of the GTI rating results in inferior diagnostic accuracy.

Reply by the author:

We changed the language for this sentence but we also feel that the Figure makes this point abundantly clear and do not feel that this needs additional explanation to support the claim.

Line 38-40: Although the ABCD2-score, the TriAGe+ score, and the PCI-score are dismissed due to lack of high-quality, prospective validation studies, the authors might want to provide some context or additional explanation for why these other scoring systems were developed and are still in use.

Reply by the author:

We added language to clarify: “Both the TriAGe+ and PCI-scores are intended for that purpose but to some extent, were designed as a “work-around” because most non-sub-specialists are not trained in evaluating subtle oculomotor findings. However, directly assessing the neurophysiology by direct physical examination will always trump epidemiological context. Application of these other scoring systems in AVS patients cannot be recommended at this time.”

Line 43-45: The comparison of the HINTS(+) and STANDING tests with MRI-DWI and brain CT is interesting, but it would be helpful to see more discussion on what this means in practical terms for clinicians and patients.

Reply by the author:

We feel that our last sentence (which has been partly re-written) addresses this point. This paragraph now reads: “Importantly, both HINTS(+) [15] and STANDING outperformed early (i.e., within first 24-48h) MRI-DWI, which has a sensitivity of 81.1 (73.3 – 88.8) and a specificity of 99.9 (99.6 – 100.0) (based on a systematic review from [13]), as shown in Figure 1. Likewise, the GTI rating outperformed brain CT [15]. This is expected since the same systematic review found a sensitivity of CT for central causes of dizziness to be less than 30% [40]. For small brainstem strokes (with a diameter of , the sensitivity of early MRI-DWI may be as low as 47%, [16] and nearly half of these strokes were due to large vessel pathology. This underscores the importance of bedside oculomotor testing in the management of acutely dizzy patients.

Line 50-56: The discussion on the challenges of diagnosing acute dizziness could be expanded to provide a more complete picture of these difficulties. For instance, discussion about how these difficulties might affect patient outcomes would be beneficial.

Reply by the author:

We added a new opening to the discussion to address this point. The first paragraph now reads: “The diagnostic work-up of the acutely dizzy patient remains challenging and absent or inadequate training in performing and interpreting subtle oculomotor findings constitute important limitations, especially in frontline providers. Most ED physicians are not yet familiar with a structured approach to the dizzy patient as e.g. outlined in the TiTrATE approach and HINTS(+) or similar bedside algorithms [18,27].”

Line 57-61: While the authors mention the potential of quantitative HINTS and the obstacles to their wider implementation, more context or examples would help underscore the importance of this point.

Reply by the author:

We did add a sentence about this, but do not think that more is needed given that it is not the main focus of the paper. It now reads: “While the introduction of quantitative (video oculography - VOG) HINTS to the ED seems promising, increasing the diagnostic accuracy beyond that of neuro-otology experts [45,46], this concept is far from being implemented broadly in routine emergency medicine practice. Significant obstacles (availability of expensive equipment and the means to interpret the findings) exist. That said, over time, VOG could become an important quality assurance tool so that frontline providers could have their results “over-read” by specialists to increase their proficiency. Validation of the findings of Wang et al [22], that the combination of a central pattern of nystagmus plus an ABCD2 score ≥4 being 100% sensitive for stroke might be effective strategy for frontline clinicians who have not mastered the HIT but are able to learn to identify central nystagmus. However, given that half of patients with cerebellar stroke do not exhibit nystagmus, we remain cautious about this approach [47].”

For a well-rounded discussion, it would be advantageous to include some recommendations for future research or suggestions for how to address the limitations and challenges identified.

Reply by the author:

In the future directions section future research directions including the implementation of quantitative HINTS in the ED setting are discussed and also suggestions are made to address current limitations in the new “conclusions” section.

Add conclusion

Reply by the author:

The former final sentence we think serves as a conclusion, so we broke it out to its own “section”

Reviewer 3 Report

The manuscript needs extensive editing of English language and sentence formation, for it to be reviewed for the content. It can be reviewed once this is done. I suggest using the help of a medical professional who is proficient in the English language and who understands the context as well.

The manuscript needs extensive editing of English language and sentence formation, for it to be reviewed for the content. It can be reviewed once this is done. I suggest using the help of a medical professional who is proficient in the English language and who understands the context as well.

Author Response

We have extensively re-written the article and the original author has taken on a well-known English speaking content expert as a co-author. We feel that the English language issues have all be clarified.

Round 2

Reviewer 2 Report

Thank you for the opportunity to review your manuscript. I wish you further success and look forward to seeing your work in print.

Reviewer 3 Report

This is a well written article comparing the various bedside clinical methods described for the evaluation of acute vestibular syndrome in the emergency setting. It summarizes well the advantages and disadvantages of these methods and points out that ED physician training and possible use of VOG may ameliorate some of these factors. 

Only minor editing required